# Vaccination-Route-Dependent Adjuvanticity of Antigen-Carrying Nanoparticles for Enhanced Vaccine Efficacy

**DOI:** 10.3390/vaccines12020125

**Published:** 2024-01-26

**Authors:** Chaojun Song, Jinwei Hu, Yutao Liu, Yi Tian, Yupu Zhu, Jiayue Xi, Minxuan Cui, Xiaolei Wang, Bao-Zhong Zhang, Li Fan, Quan Li

**Affiliations:** 1School of Life Science, Northwestern Polytechnical University, 127th Youyi West Road, Xi’an 710072, China; cj6005@nwpu.edu.cn; 2Department of Pharmaceutical Chemistry and Analysis, School of Pharmacy, Airforce Medical University, 169th Changle West Road, Xi’an 710032, China; hujinwei@fmmu.edu.cn (J.H.); liuyutao@fmmu.edu.cn (Y.L.); zhuyupupapa@fmmu.edu.cn (Y.Z.); 0518xjy@fmmu.edu.cn (J.X.); cuiminxuan2521@fmmu.edu.cn (M.C.); 3Department of Oncology, Airforce Medical Center of PLA, 30th Fu Cheng Road, Beijing 100142, China; tiannyii@163.com; 4School of Biomedical Sciences, Li Ka Shing Faculty of Medicine, The University of Hong Kong, Pokfulam, Hong Kong SAR, China; xiaoleiw@hku.hk; 5Key Laboratory of Quantitative Engineering Biology, Shenzhen Institute of Synthetic Biology, Shenzhen Institutes of Advanced Technology, Chinese Academy of Sciences, Shenzhen 518055, China; bz.zhang3@siat.ac.cn; 6Department of Physics, The Chinese University of Hong Kong, Shatin, New Territories, Hong Kong SAR, China

**Keywords:** nanoparticle adjuvant, adjuvanticity, nano-vaccine, vaccination route, mechanical property, decomposability, *Staphylococcus aureus*

## Abstract

Vaccination-route-dependent adjuvanticity was identified as being associated with the specific features of antigen-carrying nanoparticles (NPs) in the present work. Here, we demonstrated that the mechanical properties and the decomposability of NP adjuvants play key roles in determining the antigen accessibility and thus the overall vaccine efficacy in the immune system when different vaccination routes were employed. We showed that soft nano-vaccines were associated with more efficient antigen uptake when administering subcutaneous (S.C.) vaccination, while the slow decomposition of hard nano-vaccines promoted antigen uptake when intravenous (I.V.) vaccination was employed. In comparison to the clinically used aluminum (Alum) adjuvant, the NP adjuvants were found to stimulate both humoral and cellular immune responses efficiently, irrespective of the vaccination route. For vaccination via S.C. and I.V. alike, the NP-based vaccines show excellent protection for mice from *Staphylococcus aureus* (*S. aureus*) infection, and their survival rates are 100% after lethal challenge, being much superior to the clinically used Alum adjuvant.

## 1. Introduction

A vaccine requires the proper combination of pathogen-specific antigens and immunostimulating adjuvants to induce a protective adaptive immune response. NPs acting as both antigen carriers and adjuvants have been reported to improve the antigen stability and cellular uptake [1], promote the humoral and/or cell-mediated immune response [2,3,4], and thus enhance vaccine efficacy with antigen dosage reduction. For example, NPs loaded with detained toxins effectively trigger the formation of germinal centers and induce a highly neutralizing antibody titer [5].

One of the major contributions of NP-associated adjuvanticity is ascribed to the increased availability of antigens when carried by NPs. On the one hand, specific forms of antigen loading (e.g., encapsulation) in the NP carrier was found to be effective in protecting the antigen (forms such as DNA or RNA [6]) against decomposition in the biological environment [7]. On the other hand, the promotion of antigen uptake also results from the rather easy cellular uptake of NPs at both the cellular and system level, especially when a special size range of the NPs is chosen [8,9] and/or they are equipped with surface-targeting functions (e.g., to lymph nodes [10,11,12] or dendritic cells (DCs) [13,14]). A report is also available on liposome-encapsulated hen egg-white lysozyme (HEL) targeting early endosomes and thus entering the MHC class I and II presentation pathways [15]. Though NPs could increase the availability of antigens by changing the antigen loading patterns and surface targeting modification, the underlying mechanism for how the physicochemical parameters of NPs affect the antigen availability is unclarified. Moreover, different vaccination routes have different requirements regarding the NPs to enhance the antigen availability. For example, ideal vaccines for the intravenous vaccination route require better stability of the NPs in order to avoid antigen release during the circulation, as well as long circulation properties for enhancing the immune cells’ opportunities for uptake. On the other hand, in the subcutaneous vaccination route, better decomposability is needed for faster antigen release, leading to faster antigen presentation and immune response.

As a matter of fact, the effects of some of the material parameters on NPs’ adjuvanticity have been studied in the literature, but in a rather scattered manner. For example, the geometrical configuration match was identified as an effective cue to enhance the antibody binding affinity and broaden its viral neutralizing spectrum [16]. The degradability of a lipid NP carrier was observed to be a physical cue to direct antigens towards cross presentation [15]. Mechanical forces are also important determinants in immune cell activation, such as regulating cell-surface receptor activation, cell migration, intracellular signaling and intercellular communication [17,18]. More specifically, NPs that retain their force-dependent deformability were found to stimulate both humoral and cellular adaptive responses, and thus increase the survival of mice upon a lethal dose of influenza virus [18]. The stiffness of the NPs was also recognized recently as an important factor in affecting immune cell behaviors via modulating the FAK-NF-κB signaling pathway [19].

In the present work, we took *S. aureus* as a model system, due to its growing impact on public health resulting from the emergence and spread of methicillin-resistant *S. aureus* (MRSA) strains. In fact, the emergence of drug-resistant strains urged the search for alternative treatments such as immunotherapeutic approaches. Here we fabricated both soft and hard PLGA-based NPs as carriers for an *S. aureus* antigen (i.e., recombinant ess extracellular B (rEsxB)) [20] and the medium for adjuvanticity (Figure 1). We discovered that tuning the stiffness of the carrier materials by modulating the carrier composition promoted the cellular uptake of rEsxB-carrying NPs, but at the same time affected the decomposability of the NPs (and thus rEsxB release). In the mouse group receiving a S.C. vaccination, a significantly higher rEsxB-specific antibody titer was found in mice treated with a soft nano-vaccine, while similar results were obtained in mouse groups receiving a I.V. vaccination but with a hard nano-vaccine. For vaccination via S.C. and I.V. alike, the NP-based *S. aureus* vaccines show excellent protection for mice against *S. aureus* lethal challenge, and their survival rates are 100% after the inoculation of ATCC25923 strain, being far superior to the clinically used Alum adjuvant.

## 2. Materials and Methods

### 2.1. Materials

BL21 (DE3)-competent *Escherichia coli* (*E. coil*) was purchased from TIANGEN BIOTECH (BEIJING) Co., Ltd. (Beijing, China). Ni-Sepharose was purchased from Cytiva (Shanghai, China). A High-Capacity Endotoxin Removal column was purchased from Xiamen Bioendo Technology Co., Ltd. (Xiamen, China). PLGA-PEG X% polymers (X = 0, 14, 20, 25, 33, PLGA with a lactic acid to glycolic acid ratio of 50:50) and fluorescein isothiocyanate (FITC)-PEG_2000_-NH_2_ were purchased from Xi’an Ruixi Biological Technology Co., Ltd. (Xi’an, China). Isopropylbeta-D-thiogalactopyranoside (IPTG), imidazole, polyvinyl alcohol (PVA), dichloromethane (DCM), acetone, 2-(N-morpholino) ethanesulfonic acid hydrate (MES), 1-ethyl-3-(3-dimethylaminopropyl)-carbodiimide (EDC), N-Hydroxysuccinimide (NHS), bovine serum albumin (BSA), Tween 20, 2,2′-azino-bis (3-ethylbenzothiazoline-6-sulfonic acid) (ABTS), horseradish peroxidase (HRP)-labeled goat anti-mouse IgG, and 0.22 μm and 0.45 μm sterile filters were purchased from Merck (Shanghai, China). Imject™ Alum adjuvant was purchased from Thermo Fisher Scientific (Waltham, MA, USA). Ninety-six-well polystyrene plates for the enzyme-linked immunosorbent assay (ELISA) were purchased from Corning (New York, NY, USA). Enzyme-linked immunospot (ELISPOT) kits were purchased from Mabtech (Stockholm, Sweden). All other solvents are of analytical grade and were used without further purification.

### 2.2. Metheds

#### 2.2.1. Expression and Purification of rEsxB Antigen

rEsxB was chosen as the model antigen to investigate the adjuvanticity of the NP carriers. His-tagged rEsxB was prepared using the standard IPTG induction protocol [21]. Briefly, full-length wild-type EsxB DNA was subcloned into the pET-28a vector and expressed as a 6×His-tag fusion protein in BL21 (DE3) *E. coil*. rEsxB expression was induced by IPTG, purified from bacterial cell lysates by means of Ni-Sepharose chromatography, and was gradient eluted from the Ni-Sepharose with 100 mmol/L, 300 mmol/L and 500 mmol/L imidazole. The sample was then dialyzed with phosphate-buffered saline (PBS) and endotoxin was removed using the High-Capacity Endotoxin Removal column. Finally, the rEsxB solution was filtered through a 0.22 µm sterile filter and the total protein concentration was quantified using a bicinchoninic acid (BCA) protein assay. The purity and specificity of the rEsxB antigen were analyzed by means of SDS-PAGE and Western Blot.

#### 2.2.2. Preparation and Characterization of the PLGA-PEG NPs Conjugated with rEsxB

The apparent Young’s modulus of the PLGA-PEG X% (X = 0, 14, 20, 25, 33) polymers was measured using a atomic force microscope (AFM, Bruker BioScope Catalyst). Referring to the stiffness of the cell membrane [22], two sets of PLGA-PEG X% polymers (PEG_5000_-PLGA_30,000_, PEG_5000_-PLGA_15,000_) were chosen for the synthesis of PLGA-PEG X% NPs (X = 14, 25). The PLGA-PEG X% NPs were prepared via the precipitation/solvent diffusion method according to our previous work with small modifications [23,24]. Briefly, 100 mg of PLGA-PEG was dissolved in a mixture of 1.5 mL of DCM and 1 mL acetone. The polymer solution was directly added to 10 mL of a 5% PVA (Mw 31,000~50,000 Da, 87~89% hydrolyzed) solution and stirred at 600 rpm to evaporate the DCM and acetone. The mixture was then homogenized for 2 min by using a probe sonicator at 20 W to generate an oil-in-water (O/W) emulsion. The formed emulsion was added to 50 mL of deionized (DI) water and stirred for 4 h at room temperature. After that, the PLGA-PEG X% NPs were collected and filtered using a 0.45 μm filter. The NP solution was centrifuged at 16,500× *g* force at 4 °C for 1 h, washed with DI water 3 times, and lyophilized for storage.

The rEsxB was chemically linked to the PLGA-PEG X% NPs via a condensation reaction between the carboxyl group on the NPs and the amino group of the rEsxB. Then, a 10 mg sample of PLGA-PEG X% NPs was re-suspended in 10 mL of 25 mmol/L MES buffer, and 0.4 mL of 1 mol/L EDC and 0.25 mL of 1 mol/L NHS were added into the mixture and stirred for 4 h at the room temperature to activate the carboxyl groups on the NPs. After that, the NPs were centrifuged at 16,500× *g* force at 4 °C for 1 h and washed with DI water 3 times. The activated NPs were re-suspended in 1 mL of 1 mg/mL rEsxB in PBS (pH 8.0) overnight at 4 °C, then the PLGA-PEG X% NPs-rEsxB were centrifuged at 16,500× *g* force at 4 °C for 1 h, washed with DI water 3 times, and lyophilized for later use.

The hydrodynamic diameter, polydispersity index (PDI), and zeta potential of the PLGA-PEG X% NPs and PLGA-PEG X% NPs-rEsxB were measured via dynamic light scattering (DLS) (Delsa™ Nano, Beckman-Coulter, High Wycombe, UK). The successful synthesis of the PLGA-PEG X% NPs-rEsxB was determined by means of Fourier Transform infrared spectroscopy (FTIR, Thermo Nicolet IS50, Thermo Fisher Scientific, Waltham, MA, USA). The BCA protein quantification assay was used to quantify the total rEsxB concentration loaded. The loading efficiency was calculated using the following formula:Loading efficiency%=mrEsxB on the NPs (mg)mNPs (mg)×100%

#### 2.2.3. Evaluations of the Cellular Uptake and Load Release of NPs-rEsxB

Quantitative evaluation of the cellular uptake of NPs was carried out by means of the analysis of the intracellular fluorescein intensity using FITC-loaded NPs (NPs-FITC), with FITC as the florescence label. The PLGA-PEG X% NPs (X = 14, 25) were conjugated with FITC-PEG_2000_-NH_2_ via the condensation reaction between the carboxyl groups on NPs and the amino groups on FITC-PEG_2000_-NH_2_ via sulfo-NHS-catalyzed EDC activation of carboxyl groups. To simulate different administration routes of S.C. and I.V. injection, DCs (5 × 10^4^ cells/well) were fed with the 2.5 mL NPs-FITC solution at two different concentrations—2 mg/mL, for simulating a local high concentration after S.C. injection, and 10 μg/mL, for simulating a diluted environment after I.V. injection. At specific time intervals (0.5 h, 1 h, 2 h, 4 h, 6 h and 8 h), DCs were collected by means of centrifugation (250× *g* for 5 min) then re-suspended in 5 mL of RPMI 1640 medium without fetal bovine serum (FBS). The cell suspensions were frozen and thawed 4~6 times repeatedly then centrifuged at 1500× *g* for 10 min. Emission of the supernatants was measured at 528/20 nm with an excitation wavelength of 485/20 nm (Biotek SYNERGY LX Instruments, Santa Clara, CA, USA).

The load release experiment was carried out using the fluorescence method. FITC was chosen as a fluorescence tag and conjugated to the NPs in a similar manner to that of the rEsxB, then the NPs-FITC was used to examine the payload release profile of all NP samples. For the payload release studies, 50% FBS simulating the humoral environment was used. The experiments were carried out at 37 °C, and data were collected for a 12 h duration. In detail, 10 mg of PLGA-PEG X% NPs-FITC was re-suspended in 10 mL of 50% FBS at 37 °C and 1 mL of each NPs-FITC sample was collected at specific time points (1 h, 2 h, 4 h, 8 h and 12 h after release) then centrifuged at 16,500× *g* at 4 °C for 1 h. The emission of the supernatants was measured at 528/20 nm with an excitation wavelength at 485/20 nm (Biotek SYNERGY LX Instruments, Santa Clara, CA, USA).

#### 2.2.4. Evaluation of Biocompatibility of PLGA-PEG X% NPs-rEsxB In Vitro and In Vivo

We first examined the cytotoxicity of the NPs-rEsxB conjugates in vitro using fibroblast L929 cell line using the cell counting kit-8 (CCK-8) assay. Cells were seeded in a 96-well plate (5 × 10^3^ cells/well) for overnight attachment. In this process, 100 μL of the NPs-rEsxB medium solution with specific concentrations (10 ng/mL, 100 ng/mL, 1 μg/mL, 5 μg/mL, 10 μg/mL, 50 μg/mL, 100 μg/mL, 200 μg/mL, 500 μg/mL and 1 mg/mL) was added to the cell solution, respectively. Each sample was tested using 6 replicated wells. Then, the cells were incubated for another 24 h. Then,20 μL of the CCK-8 solution was added to each well and incubated at 37 °C for 1~4 h. The absorbance of cells was then measured with a microplate reader at a wavelength of 450 nm (Biotek SYNERGY LX Instruments, Santa Clara, CA, USA). The cell viabilities were calculated using the classic cytotoxicity equation:$$\mathrm{Cell}\;\mathrm{Viability}=({\mathrm{OD}}_{\mathrm{test}}-{\mathrm{OD}}_{\mathrm{background}})/({\mathrm{OD}}_{\mathrm{control}}-{\mathrm{OD}}_{\mathrm{background}})\times 100\%$$

Further evaluation of the biocompatibility of the NPs-rEsxB was carried out in vivo using BALB/c mice. All animals were monitored for their activity, physical condition, and body weight. The body weight of each mouse was measured and recorded every other day until 30 days after the first vaccination.

Hematoxylin and eosin (H&E) staining of major organs by means of the standard method published in our previous paper [25] was then carried out after the animals were sacrificed at day 42 after the first vaccination.

#### 2.2.5. Animal Immunization

All of the BALB/c mice were purchased from the Animal Experiment Center of Air Force Medical University. All animal experiments complied with the National Research Council’s Guide for the Care and Use of Laboratory Animals and the animal experiments were approved by the Animal Care and Ethic Committee of Fourth Military Medical University (Approval NO. KY20213144-1). Female 6–8-week-old BALB/c mice were divided randomly into 5 groups (n = 6 or 10 each) and immunized with PBS and rEsxB as negative control groups, clinically used Alum mixed with rEsxB (Alum-rEsxB) as the positive control group and PLGA-PEG X% NPs-rEsxB (X = 14, 25) as the nano-vaccine groups. Four groups (except PBS) were immunized either S.C. or I.V. with the same rEsxB concentration of 50 µg/mouse. To determine humoral and cell mediated response all groups were boosted with the same formulation (rEsxB concentration of 25 µg/mice) 14 and 28 days after the first immunization. Mouse serum samples were collected at specific time points.

#### 2.2.6. ELISA Assay

The rEsxB-specific antibody titer was determined through indirect ELISA on days 35 and 208 after the first immunization. Briefly, a 96-well ELISA plate was coated with rEsxB (500 ng/well) followed by blocking with 3% BSA on the next day. The plate was then washed 5 times with PBS containing 0.1% Tween 20 (PBST) and incubated with serially diluted immunized mouse serum ranging from 1:200 to 1:204,800 (100 μL/well) at 37 °C for 1 h. followed by five washes with PBST. HRP-labeled goat anti-mouse IgG (1:5000, 100 μL/well) was added and incubated at 37 °C for 45 min. This was followed by washing (as above) and development with ABTS and H_2_O_2_ as substrate chromogens. Finally, the color development for 30 min at room temperature in the dark was read at 405 nm by a microplate reader (Biotek SYNERGY LX Instruments, Santa Clara, CA, USA). The cut-off value was calculated as 2.1 times the absorbance from the negative control serum assayed [26]. In the sample group comparisons of the ELISA titer results, the area under the curve (AUC) method had similar power to the absorbance summation (AS) method, and better power than the endpoint titer (ET) method. The rEsxB-specific antibody titer results were represented by the mean serum ELISA AUC [27,28] with the S.D. of 6 mice in each group.

#### 2.2.7. ELISPOT Assay

The splenocytes of immunized animals 7 days after the third immunization (both S.C. and I.V. vaccination groups) were analyzed for IL-4 (representative of humoral immunity), IFN-γ (representative of cell-mediated immunity), IL-12 (representative of DC activation) and IL-17A (representative of TH17 T cell activation) production by means of ELISPOT assays using standard protocols [29]. The ELISPOT plates (capture antibody precoated) were washed 4 times with sterile PBS and incubated with a complete culture medium (RPMI-1640, 10% FBS) for at least 30 min at room temperature. After removing the complete culture medium, splenocytes from the immunized mice (2.5 × 10^6^ cells/well) and stimuli (4 μg/mL rEsxB) was mixed, added in the plate and incubated at 37 °C for 24 h. Phytohemagglutinin (PHA, 10 μg/mL) and the media alone served as positive and negative controls, respectively. Then, the splenocytes were removed and the plates were washed 5 times with PBS, followed by adding the detection antibodies into PBS containing 0.5% FBS (PBS-0.5% FBS) and incubation for 2 h at room temperature. The plates were washed as above and a TMB substrate solution was added to the plates until distinct spots emerged. The spots’ development was stopped by washing the plates extensively in DI water. The plates were dried in the dark at room temperature, and the spots were counted in an ELISPOT reader (Cellular Technology Limited, Cleveland, OH, USA).

#### 2.2.8. Lethal Challenge

On the 7th day after the third immunization, 100 μL of *S. aureus* (ATCC 25923) at a concentration of 1.67 × 10^8^ colony-forming units (CFU)/mL was administered to the mice in each of the groups via I.V. injection. Mice were then monitored daily for mortality and clinical signs. At the end point of the experiment, all remaining animals were euthanized.

#### 2.2.9. Serum Bactericidal Test

Standard Neisser–Wechsberg bacteriolysis evaluation [30] was employed to investigate the bacteriolysis effects of the antibody in all nano-vaccines, namely the Alum-rEsxB, rEsxB and PBS groups, with some modifications. The *S. aureus* ATCC 25923 strain was cultured until reaching the mid-exponential phase (OD_600_ = 0.5), then the *S. aureus* concentrations were adjusted to 3 × 10^6^ CFU/mL with PBS (pH 7.4). The serum of immunized mice and normal guinea pig serum were added to the *S. aureus* at a final concentration of 2% (*v*/*v*), respectively. The mixture was incubated at 37 °C for 0.5 h, plated on LB agar plates and incubated overnight at 37 °C. Finally, the number of colony-forming units of *S. aureus* was counted and the percentage of lytic bacteria was calculated using the following formula:Percentage of lytic bacteria%=1−The CFU of treated groupThe CFU of control group×100%

#### 2.2.10. Statistical Analysis

All results were expressed as mean ± standard deviation (SD) and statistical analysis was performed using the software of GraphPad Prism 9.0. Analysis of variance (students’ *t* test) was employed to determine the statistical significance of differences, and log-rank (Mantel–Cox) analysis was employed to determine the statistical significance of the survival rate. The statistical differences were defined as * *p* < 0.05, ** *p* < 0.01, *** *p* < 0.001, and **** *p* < 0.0001, and *p* < 0.05 were considered significant.

## 3. Results

### 3.1. General Characterizations of the PLGA-PEG X% NPs Conjugated with rEsxB

By manipulating the composition of the PLGA-PEG composite (i.e., PEG content), the stiffness of the material can be varied. The apparent Young’s moduli of the polymer with different PEG contents are compared in Appendix A. The higher the PEG content, the lower the apparent Young’s modulus of the composite sample. This is consistent with the literature [24]. Using the same composition recipe, we fabricated two sets of PLGA-PEG NPs with 14%, and 25% PEG (wt%), corresponding to the stiffness of ~0.18 MPa and ~0.02 MPa, respectively (Appendix A). The respective samples were named PLGA-PEG 14% NPs and PLGA-PEG 25% NPs. The stiffness of the softer sample (PLGA-PEG 25% NP) is comparable to that of the plasma membrane (~0.01 MPa) [22]. The average sizes of the NPs in both sample groups (of different PEG content) were similar (~170 nm) (Appendix A). With the PEG content increasing, the zeta potential of the NPs gradually changed from slightly negative to slightly positive (Appendix A). The purification of the rEsxB protein was confirmed by both SDS-PAGE and Western blot (Appendix A). A small increase of ~20 nm in the average size of the NPs was found after the rEsxB conjugation (Appendix A). The zeta potentials of PLGA-PEG 14% NPs-rEsxB and PLGA-PEG 25% NPs-rEsxB appeared to be slightly more negative (−2 mV~0 mV) (Appendix A). The rEsxB loading efficiencies of PLGA-PEG 14% NPs and PLGA-PEG 25% NPs were both ~3.5% (mass ratio) (Appendix A). Successful covalent binding of the rEsxB to the two groups of selected NPs was verified by the existence of amide bonds in the FTIR taken from the conjugated NPs samples (Appendix A).

### 3.2. Biocompatibility of the rEsxB Conjugated NPs

The rEsxB-conjugated NPs showed excellent biocompatibility at both the cell and animal level. Cell viability was always >95% in all the experimental groups, including rEsxB-conjugated NPs of 14 and 25 wt% PEG (Appendix A). No significant weight loss was identified in all mouse groups receiving either the S.C. (Appendix A) or the I.V. injection (Appendix A) for the administration of the NPs-rEsxB. As shown in Appendix A, no obvious lesions were observed in the H&E staining of all major organs (heart, liver, spleen, kidney and lung), irrespective of the NPs-rEsxB administration routes.

### 3.3. rEsxB-Specific Antibody

In the S.C. vaccination groups, the antibody titer was read at the background level for either the PBS- or rEsxB-treated mouse groups on days 35 and 208 after the first vaccination, respectively (Figure 2b,c). A moderate antibody level (AUC of 3365) was found at day 35 for the mouse group treated with Alum-rEsxB. The rEsxB-specific antibody for the same mouse group dropped to a barely detectable level at day 208 with an AUC of only about 332. A high rEsxB-specific antibody level in both mouse groups treated with PLGA-PEG 14% NPs-rEsxB and PLGA-PEG 25% NPs-rEsxB appeared at day 35, with an AUC of 12,857 and 66,728, respectively. Moreover, on day 208 after vaccination, the rEsxB-specific antibody for these two NPs-rEsxB-treated mouse groups maintained a relative high titer level, with an AUC of 3927 for the PLGA-PEG 14% NPs-rEsxB group and 8496 for the PLGA-PEG 25% NPs-rEsxB group.

It is worth noting that, on both day 35 and day 208 after the first vaccination, much higher antibody titer levels were found in the mouse groups treated with the softer NPs-rEsxB (PLGA-PEG 25% NPs-rEsxB) when compared to the harder NPs-rEsxB (PLGA-PEG 14% NPs-rEsxB), while both levels of the NPs-rEsxB groups were comparable to the peak level of Alum-rEsxB, even after 6 months (Figure 2b,c), when that of Alum-rEsxB dropped to a very low AUC of 332.

In the I.V. vaccination groups, superior antibody titer results were also found in the NPs-rEsxB in comparison with the Alum-rEsxB (Figure 3b,c) About 13~50 higher levels of antibody titer were found in the NP groups compared with Alum-rEsxB group on day 35 after the first vaccination. Also, longer maintenance of the antibody at a rather high level was observed in mice vaccinated with NPs-rEsxB. On day 208 after the first vaccination, a barely detectable antibody level (AUC of 5.37) could be found in the Alum-rEsxB group. However, in both NP-treated groups, the antibody titer was maintained at moderate levels, with an AUC of 4291 for the harder NPs-rEsxB and an AUC of 1965 for the softer NPs-rEsxB.

Nonetheless, an opposite trend was observed (vs. the S.C. set above) when we compared the results from the mouse groups receiving the harder or softer NPs—that is, superior antibody titer levels were found in the harder NP-treated mouse group. In particular, the PLGA-PEG 14% NPs-rEsxB demonstrated the highest level of the antibody titer among all vaccination formulations, and the peak level represented a 50-fold increase compared to that of the Alum-rEsxB on day 35 after the first vaccination. In fact, a very low antibody titer level persisted in the Alum-rEsxB using I.V. vaccination.

### 3.4. rEsxB-Specific IL-4^+^, IFN-γ^+^, IL-12^+^ and IL-17A^+^ T Cell Responses

For both the S.C. and I.V. vaccination routes, the IL-4, IFN-γ, IL-12 and IL-17A levels were always found to be much higher in the splenocyte of NPs-rEsxB-vaccinated mice than those in the splenocytes of Alum-rEsxB-, rEsxB-, or PBS-treated ones. On the other hand, the opposite trend was identified in comparing the results from the softer and the harder NPs-rEsxB groups when employing different vaccination routes—that is, the IL-4, IFN-γ, IL-12 and IL-17A levels were found to be consistently higher in the case of softer NPs (PLGA-PEG 25% NPs-rEsxB) when empoying the S.C. route (Figure 2d–g and Appendix A), while these cytokines’ levels appeared higher in the case of harder NPs (PLGA-PEG 14% NPs-rEsxB) when I.V. was adopted (Figure 3d–g and Appendix A). The opposite behavior of the softer and the harder NPs-rEsxB in cytokine secretion was consistent with that in antibody titer measurement. As IL-4 and IFN-γ serve as effective indicators for humoral and cellular immunity, respectively, the experiment results suggest that stronger immune responses (both humoral and cellular immune responses) were achieved in the NPs-rEsxB groups than in the Alum-rEsxB group. It is also interesting to point out that stronger immune responses were obtained when using softer NPs in the S.C. vaccination and harder NPs in the I.V. vaccination.

### 3.5. Lethal Challenge

Before the lethal challenge, all mice were healthy. For mouse groups treated with rEsxB or PBS, all mice died after a lethal dose of *S. aureus* challenged via intravenous injection. Overall, 30% of mice survived in the Alum-rEsxB-treated group. In comparison, the survival rates of the NPs-rEsxB groups using both vaccination routes (S.C. and I.V.) were much higher (100%) than those of the Alum-rEsxB and rEsxB groups in general (Figure 4b–g and Figure 5b–g). No significant difference was identified when using the softer or harder nano-vaccines. Moreover, no obvious body weight changes were observed in both NP-treated groups (Figure 4h and Figure 5h).

### 3.6. Neutralizing Capability of the Antibody

Bacteriolysis evaluation was employed to investigate the bacteriolysis effects of the antibody produced by means of vaccination with NPs-rEsxB or Alum-rEsxB (with rEsxB and PBS as the control groups). The results (Figure 6) reveal the significantly higher bacteria-neutralizing capability of antibodies produced by mice vaccinated with NPs-rEsxB than with Alum-rEsxB. The results are consistent with the mouse survival rates in the lethal challenge experiment.

### 3.7. Mechanism Hypothesis

The difference observed in the immune response of mice vaccinated with the softer or the harder NPs in specific vaccination routes (S.C. and I.V.) may be understood by the difference in the cellular uptake and pay load release pattern.

We first look at the cellular uptake of softer and harder NPs. For mouse groups receiving the S.C. injection, the NPs are expected to be accumulated at the injection site with a rather high concentration. A high NP feeding concentration of 2 mg/mL was chosen as one scenario for the in vitro test of cellular uptake. For mouse groups receiving the I.V. injection, the NPs will be diluted soon after entering the circulation system. When they are caught by the DCs later, the local concentration of NPs is expected to be much lower than that in the S.C. group. A low NP feeding concentration of 10 μg/mL was therefore chosen as the other scenario.

Figure 7a plots the difference in cellular uptake of the harder and the softer NPs (PLGA-PEG 14% NPs-FITC and PLGA-PEG 25% NPs-FITC) at a feeding concentration of 2 mg/mL with an incubation time from 0.5 h to 8 h. A significant difference can be found between the softer and the harder NPs. The endocytosis amount is higher when softer NPs are employed. On the other hand, when DCs were fed with a low concentration of NPs, no significant differences in cellular uptake were found when feeding the two NP samples (Figure 7b). The results reveal that when using the S.C. vaccination route, a much higher uptake amount would be associated with softer NPs when compared to the harder ones. In contrast, a similar cellular uptake amount of the NPs is expected when using the I.V. route. The higher uptake amount is associated with the higher antigen level presented by DC cells, initiating the antigen-specific immune response. This provides an explanation for the stronger immune response observed in the case of softer NPs when the S.C. vaccination route is adopted.

On the other hand, one also needs to consider that when the antigen-loaded NPs were not taken up by antigen-presenting cells in a short period of time, payload (the antigen in the present case) release from the NP carriers would take place. When this happens, physical separation between the antigen and the carrier adjuvant occurs, and the antigen is no longer bound by the carrier adjuvant, but behaves as a free antigen. This can be particularly significant when I.V. vaccination is adopted. In this regard, we observed a significantly faster release in the case of softer NPs in 50% FBS (Figure 7c). This result therefore explains the stronger immune response obtained in harder NPs compared to the softer ones when employing I.V. as the vaccination route.

## 4. Discussion and Conclusions

In order to enhance the protective efficacy of vaccines, including the induction of rapid and long-lasting immunity, new adjuvants, especially nanoadjuvants, have been studied as a promising platform with rational designs. By rationally selecting/designing polymers based on their physicochemical properties, and considering the antigen and vaccine regimen, it is possible to modulate appropriate immune responses for specific diseases.

In this study, by introducing PEG to the PLGA, we manipulated the mechanical properties (stiffness) of the NP adjuvants, which also served as carriers for the antigen of *S. aureus* (EsxB). Two typical stiffness values were chosen for preparing the NP adjuvants—PLGA-PEG 14% NPs for the hard stiffness value and PLGA-PEG 25% NPs for the soft stiffness value. Two nanovaccine systems, PLGA-PEG 14% NPs-EsxB and PLGA-PEG 25% NPs-EsxB, were then obtained for immune efficacy evaluation using two separate vaccination routes (I.V. and S.C.). As shown in the Results section, a significantly higher level of protection against *S. aureus* infection was always achieved with both hard and soft NP formulations (vs. the clinically used Alum adjuvant), as suggested by the lethal challenge experiments, with all animals being kept alive after the administration of a lethal dose of *S. aureus*. This was mainly attributed to the production of neutralized antibodies induced by NP vaccines. Also, the higher protection efficacy results of NP vaccines were consistent with the antibody titer and the cytokine analysis. Moreover, the enhanced vaccine efficacy results from improved antigen accessibility when delivered by the PLGA-PEG nanocarriers, which were also found to serve as excellent adjuvants to effectively stimulate both humoral and cellular immune responses in mice. On the other hand, a higher PEG content led to softer NPs that promoted cellular uptake, but itself was also prone to faster decomposition (and thus faster release of the antigen payload). The differences in the decomposition speed between PLGA-PEG nanocarriers may be attributed to two reasons—the molecular weight of the polymer and the existence of enzymes in the surrounding environment. The molecular weight of the PLGA chain mostly affects its degradation in indirect ways, e.g., by determining its morphology and porosity [31]. Moreover, the pancreatic lipase, which is the main component of serum lipase and subcutaneous tissue fluid, played a key role in PLGA degradation in the circulation [32,33]. The molecular weight of PLGA in PLGA-PEG 14% NPs is 30 KD, while that in PLGA-PEG 25% NPs is just 15 KD. These differences may affect the degradation of PLGA-based NPs by pancreatic lipase, so that the two PLGA-based NPs are subject to differential degradation. The interplay between these factors then determined the actual antigen uptake efficiency at the system level when different vaccination routes are utilized—that is, S.C. or I.V. injection. This provides a possible mechanism explanation for the observed higher antibody/cytokine levels induced by the rEsxB-loaded soft (hard) NPs in the case of S.C. (I.V.) vaccination. The present work provides a guideline for the vaccination-route-specific design of nanoparticle-based vaccines with maximized potency and minimized side effects.

## Figures and Tables

**Figure 1 vaccines-12-00125-f001:**
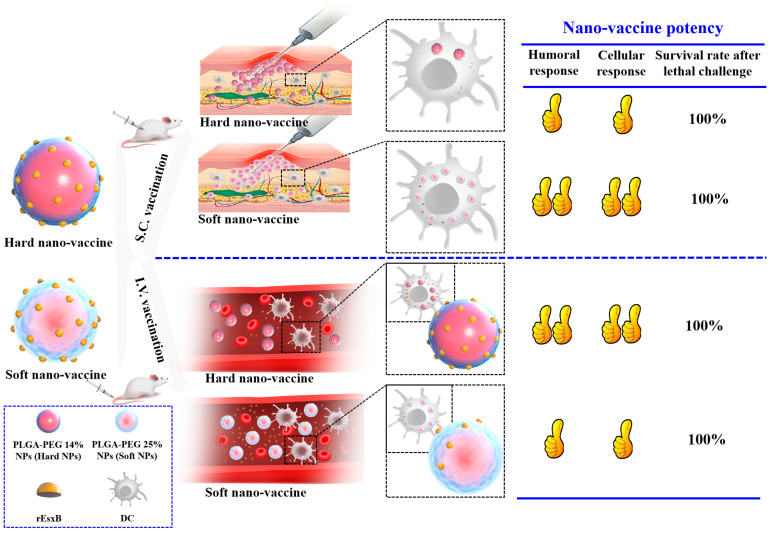
Schematic illustration of working mechanism of the developed soft and hard PLGA-based NPs as carriers for *S. aureus* rEsxB and the medium for adjuvanticity. The hard PLGA-based nano-vaccine could elicit stronger cellular and humoral immune responses through I.V. vaccination, while the soft one elicited stronger cellular and humoral immune responses via S.C. vaccination. Both nano-vaccines (either via I.V. or S.C. immunization) could protect the mice from lethal challenge of *S. aures* with survival rates of 100%.

**Figure 2 vaccines-12-00125-f002:**
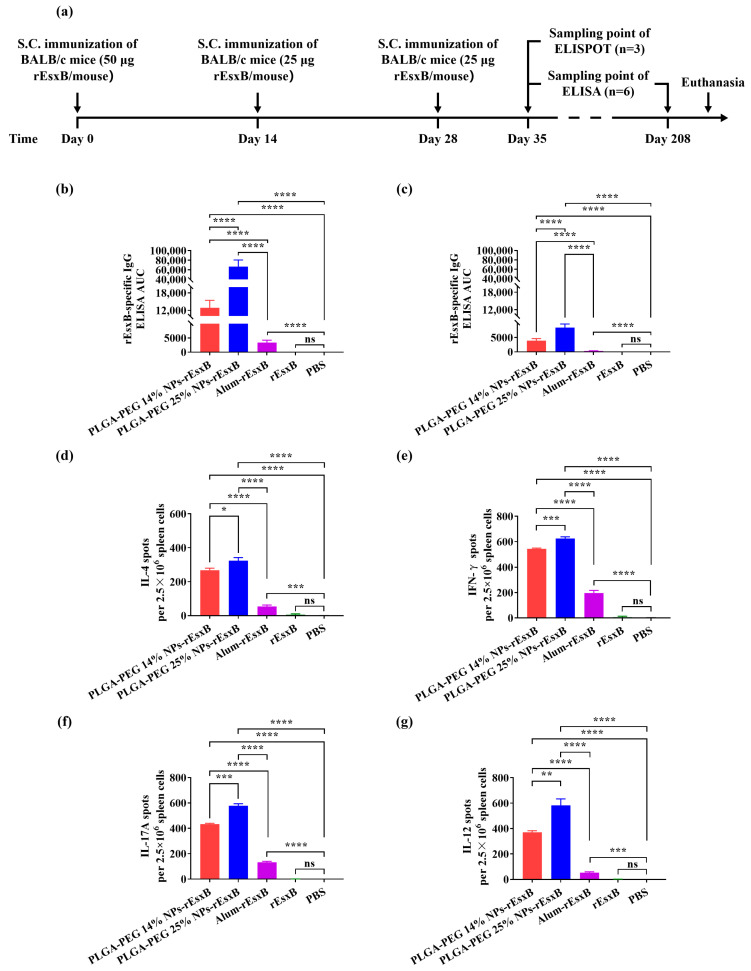
Potent humoral and cellular response to *S. aureus* vaccinations via S.C. (**a**) Illustration of the immunization protocol and the sampling time points. Serum rEsxB-specific IgG titer at (**b**) day 35 and (**c**) day 208 (n = 6). ELISpot analysis of (**d**) IL-4, (**e**) IFN-γ, (**f**) IL-12 and (**g**) IL-17A spot-forming cells among splenocytes (n = 3). All data were expressed as mean ± S.D. A *p* value < 0.05 was considered as statistically significant (* *p* < 0.05; ** *p* < 0.01; *** *p* < 0.001; **** *p* < 0.0001).

**Figure 3 vaccines-12-00125-f003:**
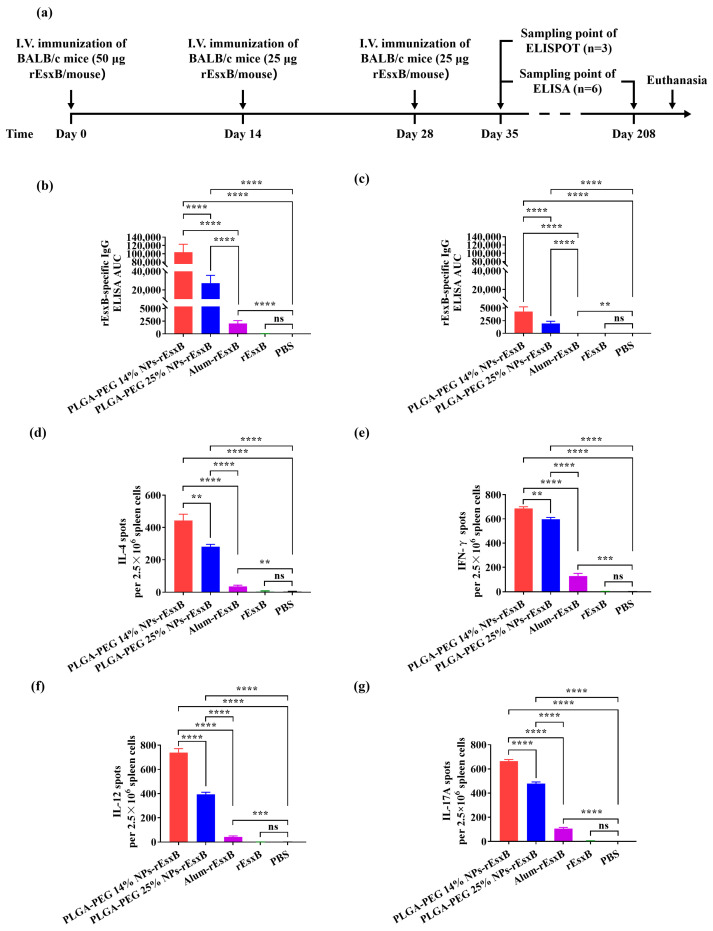
Potent humoral and cellular response to *S. aureus* vaccinations via I.V. (**a**) Illustration of the immunization protocol and the sampling time points. Serum rEsxB-specific IgG titer at (**b**) day 35 and (**c**) day 208 (n = 6). ELISpot analysis of (**d**) IL-4, (**e**) IFN-γ, (**f**) IL-12 and (**g**) IL-17A spot-forming cells among splenocytes (n = 3). All data were expressed as the means ± S.D. A *p* value < 0.05 was considered as statistically significant (** *p* < 0.01; *** *p* < 0.001; **** *p* < 0.0001).

**Figure 4 vaccines-12-00125-f004:**
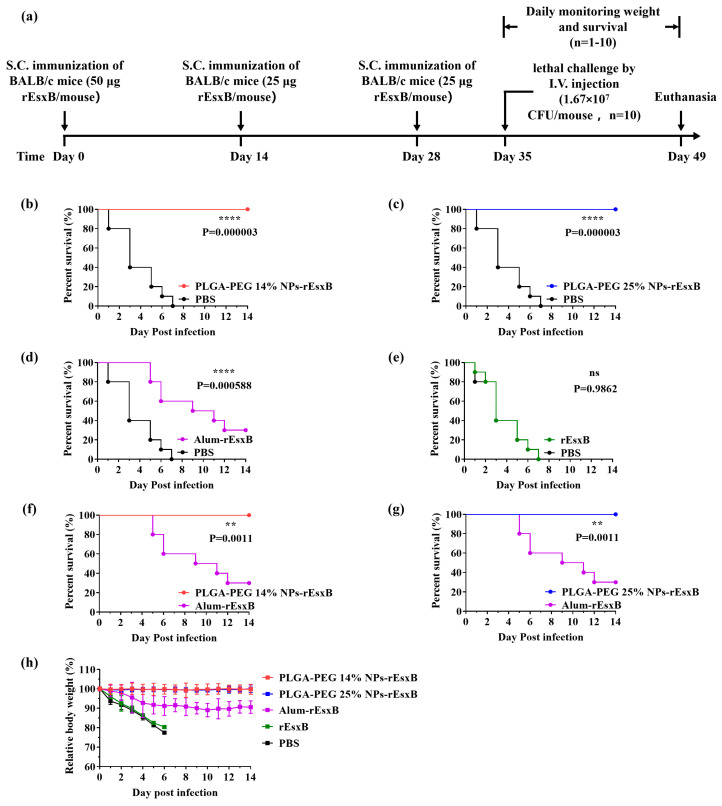
Survival rate comparisons on day 14 after lethal challenge of *S. aureus* in the S.C. vaccination groups. (**a**) Timeline for immunization, challenge and evaluation of protective efficacy. (**b**–**g**) Survival rate (n = 10) and (**h**) weight loss (n = 1–10) of the immunized mice. Survival rates were analyzed with Log-rank (Mantel–Cox) analysis, and mouse weight is shown as the mean ± S.D. A *p* value < 0.05 was considered as statistically significant (** *p* < 0.01; **** *p* < 0.0001).

**Figure 5 vaccines-12-00125-f005:**
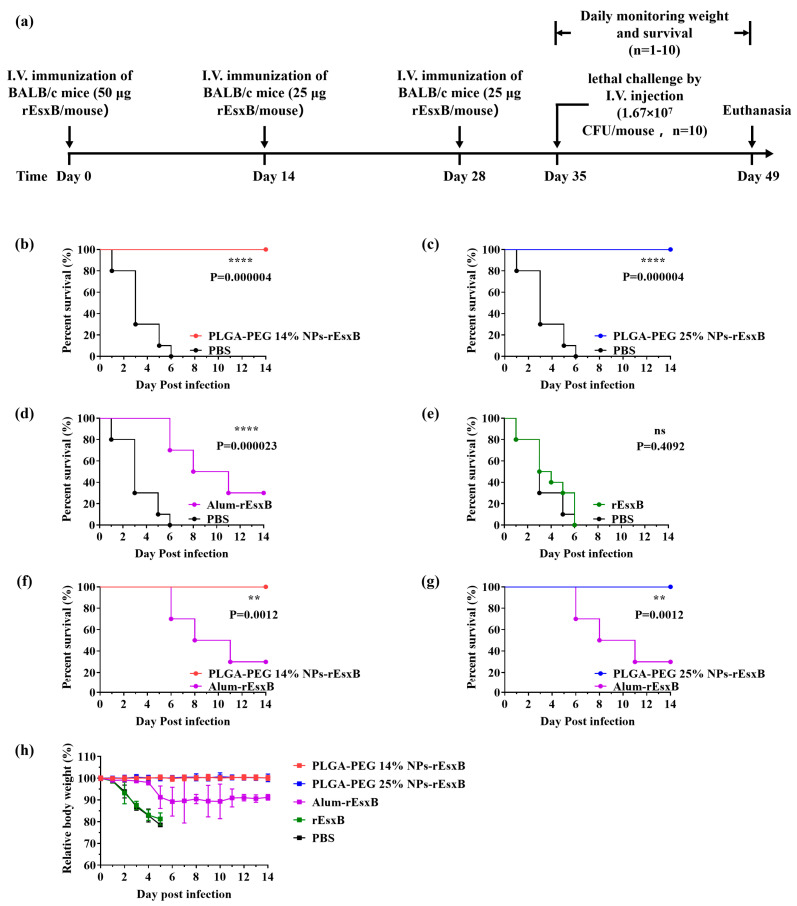
Survival rate comparisons on day 14 after lethal challenge of *S. aureus* in the I.V. vaccination groups. (**a**) Timeline for immunization, challenge and evaluation of protective efficacy. (**b**–**g**) Survival rate (n = 10) and (**h**) weight loss (n = 1–10) of the immunized mice. Survival rates were analysed with Log-rank (Mantel–Cox) analysis, and mouse weight is shown as the mean ± S.D. A *p* value < 0.05 was considered as statistically significant (** *p* < 0.01; **** *p* < 0.0001).

**Figure 6 vaccines-12-00125-f006:**
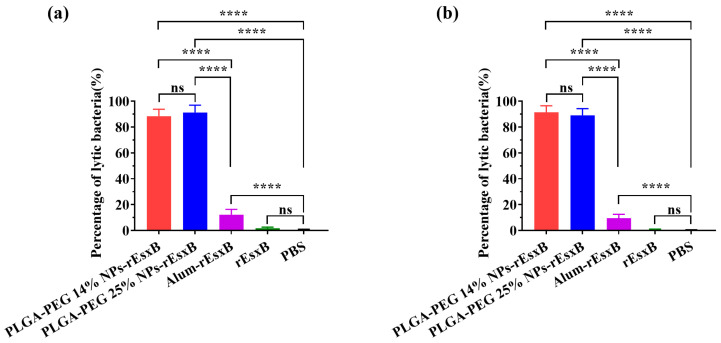
The bacteriolysis of serum antibody by (**a**) S.C. and (**b**) I.V. (n = 6). All data were expressed as the means ± S.D. A *p* value < 0.05 was considered as statistically significant (**** *p* < 0.0001).

**Figure 7 vaccines-12-00125-f007:**
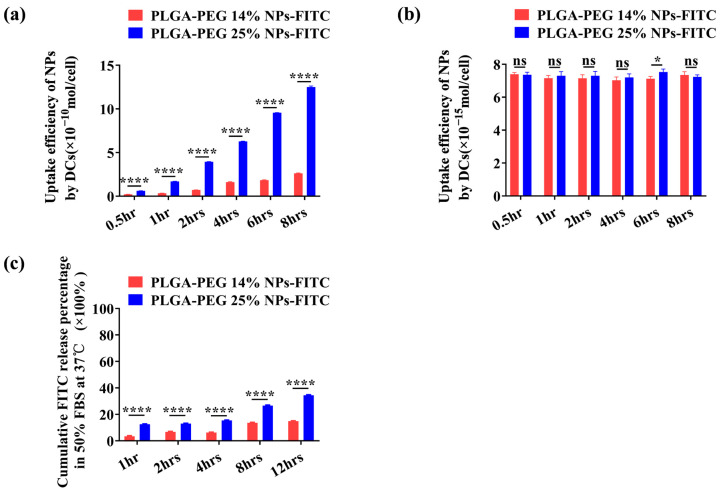
Cellular uptake efficiency of PLGA-PEG X% NPs-FITC at (**a**) a high feeding concentration of 2 mg/mL and (**b**) a low feeding concentration of 10 μg/mL. (**c**) In vitro cumulative FITC release behavior of PLGA-PEG X% NPs-FITC was studied in 50% FBS at 37 °C. The release of FITC was quantified in triplicate wells for each group. All data are expressed as the means ± S.D. *p* values < 0.05 were considered as statistically significant (* *p* < 0.05; **** *p* < 0.0001).

## Data Availability

Data will be made available upon request.

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
