# Peer review of "Vaccination-Route-Dependent Adjuvanticity of Antigen-Carrying Nanoparticles for Enhanced Vaccine Efficacy"

_vaccines, 2024, doi:10.3390/vaccines12020125_

Round 1

Reviewer 1 Report (Previous Reviewer 3)

Comments and Suggestions for Authors

The authors have addressed all my comments from their previous submission of this manuscript and suitably improved this new version.

Author Response

Comments: The authors have addressed all my comments from their previous submission of this manuscript and suitably improved this new version.

Response: Thanks for your valuable contributions to our work.

Reviewer 2 Report (Previous Reviewer 1)

Comments and Suggestions for Authors

No additional comments or suggestions.

Comments on the Quality of English Language

Minor editing of English language required

Author Response

Comments: Minor editing of English language required

Response: We had checked the manuscript word by word and all the changes were marked in the revised version.

Reviewer 3 Report (New Reviewer)

Comments and Suggestions for Authors

The submitted manuscript presents a highly original approach for intravenous and subcutaneous vaccination using conjugates composed of synthetic mixed polymeric nanoparticles consisting of different amounts of the biodegradable polylactic-co-glycolic acid co-polymer (PLGA) and and polyethyleneglycol (PEG). Using the staphylococcal virulence factor EsxB protein as a test antigen, the authors could show that nanoparticles with a larger amount of PLGA show a higher Young's modulus, greater softness as well as a stronger adjuvanticity in subcutaneous vaccination. Nanoparticles with a lower percentage of PLGA in the composite polymer showed a lower Young’s modulus and demonstrated a stronger adjuvanticity in intravenous vaccination. Excitingly, the authors could show that vaccination with EsxB-loaded nanoparticle resulted in a survival rate of 100% and was therefore much superior to the conventional aluminum adjuvant in protecting BALB/c mice from lethal challenge with S. aureus ATCC25923 strain.

I strongly recommend this original work for publication and have only some minor suggestions for improvement.

Minor:

-          All abbreviations should be explained when first used in the text. Please check.

-           

-          Methods section: An unusual and not very common absorbance summation method was used for the ELISA read out. Has this been done according to this publication? : Hartman H, Wang Y, Schroeder HW Jr, Cui X. Absorbance summation: A novel approach for analyzing high-throughput ELISA data in the absence of a standard. PLoS One. 2018 Jun 8;13(6):e0198528. doi: 10.1371/journal.pone.0198528. PMID: 29883460; PMCID: PMC5993274.? If so, please cite this as are reference and also explain why this read out was chosen over the common standard ELISA read out by sigmoidal curve fitting.

-           

-          Figure 1: The figure legend should be extended to include much more details so that the figure could be read without reference to the text.

  • Figure 1: resolution appears to be inferior to publication standards Raise DPI for lines to at least 600 DPI and colour items to at least 300 DPI resolution. If the image remains at low quality even at elevated resolution the figure should be redrawn using a standard font and care should be taken to ensure the font is embedded so as to avoid pixelated text.
  •  
  • Discussion: The authors have limited their interpretation of the results pertaining to the observed differences in subcutaneous and intravenous vaccination to the degree of softness of the nanoparticle-EsxB conjugates. Considering the fact that the different nanoparticles are chemically similar and the EsxB antigen has been conjugated covalently to both types of particles, it seems to be interesting whether the nanoparticles are subject to differential degradation by hydrolytic enzymes such as esterases and lipases and whether such enzymes are present in the subcutaneous interstitial fluid and/or in the blood. This would be an interesting addition to the discussion.
Comments on the Quality of English Language

While the English writing is ok and legible, there is still some room for additional improvement: For example:

“underneath mechanism”à better: underlying mechanism; “fabricated”à unfortunate word with ambiguous /double meaning:àbetter: manufactured; and other grammar and style issues etc.

Given the excellent data and highly interesting and important results I suggest to invest in a professional proofreading/editing service by a native speaker.

Author Response

This manuscript is a resubmission of an earlier submission. The following is a list of the peer review reports and author responses from that submission.

Round 1

Reviewer 1 Report

Comments and Suggestions for Authors

Lines 57-59  Please clarify

Nonetheless, how do various material parameters of the NPs affect the availability of the loaded antigen is less discussed, let along their dependence on the vaccination routes, which is important for practical application. 

Line 103 Thermo Fisher Scientific

Line 188 Cell viability  equation  - add x100 

Comments on the Quality of English Language

Lines 57-59 Nonetheless, how do various material parameters of the NPs affect the availability of the loaded antigen is less discussed, let along their dependence on the vaccination routes, which is important for practical application. 

Reviewer 2 Report

Comments and Suggestions for Authors

I thank the authors for their research. The authors loaded two types of NPs and studied the immune response upon immunization with these particles.

Unfortunately, the work lacks a number of important controls: there is no data for unloaded NPs. The study used two methods of immunization of NPs (intravenous and subcutaneous), but differences in the immune response were not shown for each immunization method. In addition, the authors used different concentrations of NPs for intravenous and subcutaneous administration, which does not allow comparison of these methods with each other.

Surprisingly, despite the low loading efficiency (~3.5%) of NPs, immunization with loaded particles resulted in a significant immune response. And immunization with a “naked” antigen does not lead to the development of at least some antibody titer.

Reviewer 3 Report

Comments and Suggestions for Authors

Review of manuscript # 2646028 for Vaccines

This manuscript studies the use of antigen-carrying nanoparticles as potential vaccines using S. aureus as a model system. PGLA/PEG nanoparticles of different physical strength are produced by varying PEG content and coated with rEsxB as an antigen. The function of these is directly compared to Alum, with equivalent antigen content, in both I.V and S.C routes of administration.

Both B and T-cell responses are determined and ability to provide protection against lethal challenge in mouse models.

Biophysical characterization shows the nanoparticles to be of similar size and degree of antigen loading. Varying the PEG content results in nanoparticles which are defined as “soft” and “hard”.

In all situations both types of nanoparticles significantly outperform Alum.

Interestingly stronger immune responses were seen using soft NPs when administered S.C but the opposite was seen when administering I.V. The authors provide a plausible explanation for this in their discussion.

In general, the studies are well described, use appropriate controls and the data clearly presented. As the authors state, this work provides a guideline for future research on vaccination-route-specific design of nanoparticle-based vaccines.

In my opinion the manuscript is suitable for publication in Vaccines, with the few following modifications.

1.       Figure S2 does not seem to be referred to in the text. This should be corrected, or it removed.

2.       Generally, the labelling of figure axis is too small, making figure difficult to read without magnification. New legible figure should be produced.

3.       The English language usage is in need of minor improvement, especially in the correct case usage.

Comments on the Quality of English Language
